# Mycotoxin Occurrence in Feeds and Raw Materials in China: A Five-Year Investigation

**DOI:** 10.3390/toxins15010063

**Published:** 2023-01-11

**Authors:** Wei Hao, Shu Guan, Anping Li, Jinyong Wang, Gang An, Ursula Hofstetter, Gerd Schatzmayr

**Affiliations:** 1Department of Animal Nutrition and Health, DSM (China) Co., Ltd., Shanghai 201203, China; 2Department of Animal Nutrition and Health, DSM Singapore Industrial Pte. Ltd., Singapore 117440, Singapore; 3Department of Animal Nutrition and Health, DSM Austria GmbH, 3131 Getzersdorf, Austria

**Keywords:** mycotoxin occurrence, new-season corn, feed, raw material, China

## Abstract

Mycotoxins are ubiquitously present in feeds and raw materials and can exert toxicity on animals and humans. Therefore, mycotoxin occurrence should be monitored. We report here a multi-mycotoxin survey of feed samples in China from 2017 to 2021. Concentrations of aflatoxins, trichothecenes type B, fumonisins, and zearalenone were determined in a total of 9392 samples collected throughout China. Regional differences and year-to-year variation of mycotoxin occurrence were also assessed in new-season corn. Generally, *Fusarium* mycotoxins were prevalent, while mycotoxin contamination in each feed commodity showed a distinct pattern, e.g., wheat and bran were typically affected by trichothecenes type B, peanut meals were highly susceptible to aflatoxins, and finished feeds exhibited a comparatively high prevalence of all mycotoxins. In new-season corn, trichothecenes type B and fumonisins were most prevalent, with positive rates of 84.04% and 87.16%, respectively. Regions exhibited different patterns of mycotoxin occurrence. The Anhui and Jiangsu provinces of East China exhibited a high prevalence and concentrations of aflatoxins with a positive rate and a positive average of 82.61% and 103.08 μg/kg, respectively. Central China obtained high fumonisins levels of 4707.84 μg/kg. Trichothecenes type B and zearalenone occurred more frequently in temperate regions of Northeast China, and their positive rates reached 94.99% and 55.67%, respectively. In these regions, mycotoxin concentrations in new-season corn exhibited pronounced year-to-year variations and this could be due to the unusual changes of rainfall or temperature during sensitive periods of corn growing. A large fraction of new-season corn samples contained multiple mycotoxins with two to three classes (75.42%), and the most frequently observed co-contaminants were the combination of trichothecenes type B and fumonisins (73.52%). Trichothecenes type B and zearalenone concentrations were highly positively correlated with a coefficient of 0.775. In conclusion, mycotoxins contamination and co-contamination of feeds are common. Mycotoxin contamination in new-season corn exhibited regional patterns and year-to-year variations, with climate and weather conditions as determinant factors.

## 1. Introduction

Recently, as the demand lifted the utilization of crop by-products as a new source of feed materials with the development of the livestock industry in China, the public concern for food and feed safety, especially for mycotoxin contamination, has increased and become an important issue to be addressed. Contamination levels of certain types of mycotoxins are generally considered as the hygienic indicators in the food and feed chains [1], and these mycotoxins are well known to exert toxicity on animals, compromising the safety and health of humans through the carryover effects from animal products to human consumers [2,3].

Mycotoxins are toxic fungal secondary metabolites, and statistically more than 300 are known nowadays [4]. The most important genera-secreting mycotoxins are *Aspergillus*, *Fusarium*, and *Penicillium*, and the most prevailing mycotoxins produced by the above-mentioned fungi genera are aflatoxins (e.g., aflatoxin B1; AFB1), fumonisins (FUMs), zearalenone (ZEN), trichothecenes type B (e.g., deoxynivalenol; DON), trichothecenes type A (e.g., T-2 toxin; T-2), and ochratoxin A (OTA). The fungal infestation and mycotoxin accumulation could occur widely and frequently in the crop field with climate conditions of rainfall and weather temperature as the most influential factors [3], and since the same fungal strain could produce different types of mycotoxins even under similar environmental conditions, mycotoxins frequently co-occur in feeds and materials.

For mycotoxin risk management, regulatory limits for mycotoxin values have been implemented in many regions throughout the world. In the European Union, maximum levels are enforced for AFB1 and guidance values are stipulated for FUMs, DON, ZEN, and OTA [5,6]. In China, the allowable maximums are regulated for the concentrations of AFB1, ZEN, DON, FUMs, T-2, and OTA in finished feeds and plant materials, and according to different livestock species, the levels for these regulated mycotoxins are also set with different limits (Table 1) [7].

With the legislation put in place, the mycotoxin contamination in feeds has been alleviated and most of the feed samples could meet the requirements of the maximum limits of mycotoxin in the relevant regulations in China, even though continuous low or medium concentration exposure to animals could lead to chronic toxicity. Furthermore, since mycotoxins co-occur in most cases, the effect of the co-contaminants could be “additive” or “synergistic” when compared to the effects of individual mycotoxins, and these interactions may be produced even though the single mycotoxins are at relatively low levels [8,9], while available data are still scarce and insufficient.

The analysis of mycotoxin occurrence is of particular interest and draws increasing attention. Considering that the environmental conditions of temperature and precipitation are key determinants of mycotoxin contamination [10], such survey would exhibit specific pollution patterns in different geographic regions [11], as well as support the mycotoxin risk warning management due to the changing climatic conditions or specific meteorological events. Most importantly, through the monitoring and supervision of mycotoxin contamination in feeds, these surveys would help with the reduction in intake of different types of mycotoxins and the alleviation of acute or chronic symptoms of toxicity in animals, ensuring body health and minimizing the economic damage in the livestock industry.

Previous surveys have mostly reported the regulated mycotoxins and their co-occurrence either on a global scale [11,12,13] or with covers of smaller geographic regions, such as African, European, and Asian countries [14]. In China, the surveys focusing on the patterns of mycotoxin contamination in different commodities and in different regions have been published [15], and with the increased awareness, the demand to investigate the pollution characteristics during relatively long years has become more urgent. Thus, in this report, we analyzed the occurrence and co-occurrence of aflatoxins, ZEN, trichothecenes type B, and FUMs in 9392 samples of finished feed and feed raw materials such as corn, wheat, soybean, and corn by-products collected throughout China from 2017 to 2021. The mycotoxin occurrence in new-season corn of five geographic regions was compared and the year-to-year variation of mycotoxin concentrations was also analyzed. The relevant statistics and reports are still scarce in China, and so far, this is the first and the most comprehensive investigation of regional and year-to-year trends of mycotoxin occurrence in China, with the largest dataset included in this report.

## 2. Results

### 2.1. General Mycotoxin Occurrence

The numbers of tested samples, the respective positive rates, as well as the average positives and maximums of different types of mycotoxins are presented in Table 2. A total of 9329 samples were collected throughout China and analyzed for the occurrence of aflatoxins, trichothecenes type B, FUMs, and ZEN. Generally, the *Fusarium* mycotoxins of trichothecenes type B, FUMs, and ZEN were prevalent and were detected in 87.07%, 77.14%, and 56.29%, respectively, of the total samples, with positive averages of 838.89 μg/kg, 1842.57 μg/kg, and 183.81 μg/kg, respectively. In the year 2017, aflatoxins contamination occurred in 12.71% of the investigated samples with a positive average of as high as 87.15 μg/kg. The positive rates of trichothecenes type B and FUMs in that year also exhibited high values of 98.74% and 71.07%, respectively. In the year 2018, aflatoxins remained at a relatively high pollution level with a positive rate of 25.25% and an average of positives of 81.13 μg/kg. FUMs also exhibited a high prevalence of 80.60%, and its level of positive averages reached 2464.74 μg/kg. Aflatoxins contamination was significantly alleviated from 2019 to 2021 (*p* < 0.01). The positive rates remained below 20%, and the averages of positives were at levels of approximately or below 30 μg/kg in this study. In the year 2019, ZEN exhibited a low positive average of 74.20 μg/kg. In the year 2020, trichothecenes type B, FUMs, and ZEN were detected in 72.44%, 62.91%, and 46.82% of the samples, respectively, and the positive rates were comparably lower than those in other years. In the year 2021, a higher prevalence of 79.09% and a higher positive average of 1135.07 μg/kg were determined in ZEN and trichothecenes type B (*p* < 0.01), respectively.

### 2.2. Mycotoxin Occurrence in Different Feed Commodities

Mycotoxin occurrence was compared in different feed commodities in this survey and the results are presented in Table 3. Mycotoxin occurrence differed according to different feed commodities.

In corn, FUMs and ZEN exhibited a higher prevalence of 81.55% and 51.51% and higher positive averages of 2618.81 μg/kg and 176.79 μg/kg, respectively. Corn also obtained a relatively high prevalence of trichothecenes type B (87.47%), and the positive rate of aflatoxins was below 20%. With a similar mycotoxin occurrence than for the corn samples, corn by-products, which are dried distillers’ grains and soluble fraction (DDGS) and corn gluten meal in this study, were concentrated in ZEN and FUMs contamination. DDGS exhibited a high prevalence of *Fusarium* mycotoxins with the positive rates of 98.48%, 86.80%, and 87.82%, respectively, in trichothecenes type B, FUMs, and ZEN, and the average of positives reached a value as high as 2327.08 μg/kg in trichothecenes type B. Contaminations of FUMs and ZEN were even aggravated in samples of corn gluten meal with positive averages of 8363.50 μg/kg and 2004.30 μg/kg, respectively. The average of positives of trichothecenes type B was slightly lower in samples of corn gluten meal than in corn samples.

In wheat, bran, and rice bran meal, trichothecenes type B was the most frequently detected mycotoxin, especially in wheat samples, wherein the positive average reached 2129.29 μg/kg in this survey. In soybean meal, ZEN was prevalent and detected with a positive rate of approximately 40%. Cottonseed meal was mainly polluted by the infestation of aflatoxins and FUMs. In the case of peanut meal samples, the contamination of aflatoxins was determined at a high risk level, with a positive rate of 100.00% and a positive average of up to 417.72 μg/kg.

*Fusarium* mycotoxins dominated in silage, while the average concentrations of positives in FUMs were markedly lower than those in corn. Aflatoxins were rarely detected in alfalfa and oat. The most prevalent mycotoxin was trichothecenes type B with positive rates of above 50% in both grasses, and the positive average of trichothecenes type B should be paid more attention to in oat, where its value reached 1728.82 μg/kg.

Swine and poultry finished feeds were among the commodities exhibiting relatively high percentages of positive samples for every mycotoxin analyzed in this survey. The concentrations of trichothecenes type B reached positive averages of above 500 μg/kg in both finished feeds, and FUMs were detected at positive averages of 120.13 μg/kg and 83.39 μg/kg in poultry and swine feeds, respectively. In ruminant feeds of concentrate supplement and TMR, aflatoxins showed lower prevalence and positive averages than in other commodities, and 53.14% and 71.91% of samples of concentrate supplement and TMR were contaminated with ZEN, respectively.

### 2.3. Mycotoxin Occurrence in New-Season Corn

Mycotoxin occurrence in new-season corn from 2017 to 2021 were summarized in Table 4. Trichothecenes type B and FUMs were most prevalent, with positive rates of 84.04% and 87.16%, respectively, followed by ZEN, (50.13% positive rate), and aflatoxins positive samples were less common, with a percentage of 31.40% in this survey. The level of aflatoxin contamination exhibited a positive average of 63.83 μg/kg and the maximum value reached 733 μg/kg, which was sourced in a sample collected in the year 2018.

In the year 2017, the average of positives for aflatoxins was higher than in the following years with the value of 147.74 μg/kg (*p* < 0.01), and in the year 2019, the positive samples exhibited much lower contamination levels of aflatoxins, FUMs, and ZEN with positive averages of 10.46 μg/kg, 1944.99 μg/kg, and 62.88 μg/kg, respectively. In the year 2021, the contamination of trichothecenes type B and ZEN exhibited highest positive averages of 1285.93 μg/kg and 300.73 μg/kg, respectively, when compared to previous years (*p* < 0.01). These values were high enough to draw the public attention.

### 2.4. Regional Variation of Mycotoxin Occurrence in New-Season Corn

The regional variation of mycotoxin contamination in new-season corn was presented in Table 5 and the year-to-year variations of mycotoxin concentrations in different regions were shown in Figure 1. For Northeast China of Gansu province, the year-to-year variation of mycotoxins concentrations was not evaluated as the low sample number did not allow for this analysis. The results for each region of Northeast China (Heilongjiang, Jilin, and Liaoning provinces), North China (Hebei and Inner Mongolia provinces), East China (Shandong province), East China (Anhui and Jiangsu provinces), and Central China (Henan China) are described in detail as follows:

#### 2.4.1. Northeast China

In samples from Northeast China, trichothecenes type B and ZEN were at the top prevalence ratio with 94.99% and 55.67%, respectively, in comparison with samples from other regions. The contamination levels of trichothecenes type B increased from 2019 and peaked in 2021 with a positive average of 1317.88 μg/kg. The prevalence of ZEN gradually increased from 2017 to 2021 in this region. In addition to trichothecenes type B and ZEN, FUMs were also prevalent in this sub dataset with 79.42% of positive samples, but the prevalence and contamination level of the positive average were lower than any other regions of China (*p* < 0.01). Aflatoxins were detected in the lowest fraction of samples (3.69%), and at the lowest positive average of 9.36 μg/kg (*p* < 0.01). The positive ratios and contamination levels of FUMs both peaked in samples from 2018.

#### 2.4.2. North China

In North China, FUM was most frequently detected with an 89.95% positive rate, and its contamination peaked with a 100.00% positive rate and a high average concentration of 7016.97 μg/kg in the year 2018. The prevalence of trichothecenes type B was also at a relatively high level (80.90%), and 49.75% of the samples were detected as positive in terms of ZEN contamination in this region. Aflatoxins were detected in a positive fraction of 30.15% and at a relatively low average concentration of 15.36 μg/kg (*p* < 0.01).

#### 2.4.3. East China (Shandong)

In Shandong province of East China, FUM was most dominating (90.05%. positive rate) reaching the highest level of positive average of 4699.88 μg/kg (*p* < 0.01). Trichothecenes type B and ZEN also exhibited a high prevalence of 76.96% and 59.16%, respectively, with averages of positives of 1207.20 µg/kg and 483.87 µg/kg, respectively, which were both the highest concentration obtained for any region investigated in this survey (*p* < 0.01, *p* < 0.01). Furthermore, the prevalence and levels of both trichothecenes type B and ZEN significantly increased from 2019 to 2021. Their positive averages were particularly high, and the concentrations of 1954.00 μg/kg and 722.00 μg/kg were detected, respectively, in trichothecenes type B and ZEN in the year 2021. Aflatoxins were detected in a low fraction of the samples (8.38%) and with a low positive average level (35.51 μg/kg) (*p* < 0.01), and the contamination level was higher in 2018 than in other years.

#### 2.4.4. East China (Anhui and Jiangsu)

Aflatoxins were detected in 82.61% of samples from the Anhui and Jiangsu provinces, which was the highest percentage in all the invested regions, reaching a particularly higher average concentration of 103.08 µg/kg when compared to other regions (*p* < 0.01). Accordingly, high fractions and high levels were detected throughout the 5-year period, except in the year 2019, when aflatoxins were detected at a relatively low level with a positive average of 19.00 μg/kg, and even the prevalence remained at a high fraction of positive samples. FUM was most prevalent (92.03% positive rate) at a highest level of positive average of 4606.60 μg/kg (*p* < 0.01). The contamination of trichothecenes type B (*p* < 0.01) and ZEN (*p* < 0.01) was detected at the lowest levels relative to other regions in this survey.

#### 2.4.5. Central China

FUM was the most frequently detected mycotoxin with 91.87% of positive samples, reaching the highest level with a positive average of 4707.84 μg/kg (*p* < 0.01). FUM concentrations were particularly high in the year 2018 and then tended to decrease between 2019 and 2021. Aflatoxins were detected in 64.23% of samples, with average concentrations of 61.62 μg/kg. Trichothecenes type B and ZEN were detected with positive rates of 82.52% and 43.09%, respectively.

### 2.5. Co-Occurrence of Mycotoxins in New-Season Corn

In order to analyze the co-occurrence of mycotoxins in new-season corn, the proportions of samples contaminated with either combination of mycotoxins were calculated in this study (Figure 2). Of all the samples, 75.42% contained two to three classes of mycotoxins, and 9.09% of the samples were contaminated with all four mycotoxins. The co-occurrence of trichothecenes type B and FUMs was most frequently observed, with the ratio of 73.52%. Trichothecenes type B and ZEN as well as ZEN and FUMs co-occurred in 42.67% and 39.45% of samples, respectively. The combination of three mycotoxins of ZEN, trichothecenes type B, and FUMs was detected in 39.06% of the samples.

In Figure 3, the combination of any two mycotoxins in new-season corn in the 5-year period were presented. The concentrations of trichothecenes type B and ZEN exhibited a positive correlation and their coefficient reached 0.775 (*p* < 0.0001). The concentrations of aflatoxins and FUMs also exhibited a positive correlation with a coefficient of 0.199 (*p* < 0.0001). Other mycotoxin combinations exhibited negative correlation coefficients in this survey.

## 3. Discussion

Mycotoxin contamination in feeds and raw materials was very common. In general, aflatoxins showed a low prevalence, while the *Fusarium* mycotoxins of FUMs and trichothecenes type B were more frequently detected, followed by ZEN. Even though there are only limited survey data available, these results can be supported by previously published studies. In one study conducted in China, mainly in the first half of 2009, the frequency of samples positive for ZEN, DON, and FUMs reached as high as 97.60%, 95.20%, and 82.70%, respectively [1]. In another 10-year mycotoxin survey on a global basis, FUMs, DON, and ZEN were investigated with 60.7%, 84.8%, and 58.2% of samples in the East Asian sub-dataset, and DON was even more prevalent than in samples from other regions worldwide [11]. With the high average contamination levels detected in *Fusarium* mycotoxins in this survey, we should see that the high-risk of trichothecenes type B, ZEN, and FUMs in feeds is a recurring issue in China and should draw serious attention. According to Gruber-Dorninger et al. [11], AFB1 was detected in 17.1% of samples from East Asia at a positive average concentration of 10 μg/kg, and the occurrence pattern of aflatoxins was similar in this survey, while the contamination level was much higher. From 2017 to 2021, with the increasing awareness of the aflatoxins’ toxicity, the aflatoxins risk has been alleviated gradually. The improved agriculture practices and grain storage conditions, as well as the rational application of aflatoxins detoxifier may have contributed to this significant shift in mycotoxin risk.

### 3.1. Occurrence of Mycotoxins in Different Feeds and Raw Materials

Aflatoxins are mainly produced by *Aspergillus flavus* or parasitic *Aspergillus*, and these species easily infect crops with high oil content [16], such as corn, peanut, and cottonseed [17]. The positive rate of aflatoxins in corn was found to be 16.50% with a relatively high positive average concentration of 63.28 μg/kg in this survey, while in other studies, the occurrence pattern may be varied due to the different sampling distribution or determination approaches. In a survey conducted from 2016 to 2017, the positive detection rate for AFB1 in corn collected from 21 provinces were 96.1% and the averages of positives were 5.8 μg/kg, and 4.1 μg/kg, in 2016 and 2017, respectively [18]. Meanwhile, in another study with 44 corn grains collected from three major corn-producing provinces in North China (Shandong, Hebei, and Henan) in 2014, aflatoxins exhibited 2.27% of positive samples and as high as 148.4 μg/kg of positive average concentration, indicating that aflatoxins are posing significant risks to corn in China, in terms of both prevalence and exposure level. Peanuts are also susceptible to aflatoxin contamination. A two-year survey conducted in the main peanut-producing regions of China has reported 95% of positive samples but mainly at low levels [19]. Peanut meal is the product resulting from the extraction of peanut oil, and aflatoxins may be highly enriched during this process. In the latest surveys of the years 2021 and 2020 scaled in China, aflatoxins were detected positively in all peanut meal samples with average concentrations of more than 100 μg/kg, which reached the highest values among all the investigated feed commodities in both years. In DDGS and corn gluten meal, as the corn by-products, the contamination of aflatoxins was also highly prevalent, which should be paid special attention to.

FUM is a type of mycotoxins produced primarily by the fungi *Fusarium verticillioides* and *F. proliferatum*, and these species mainly contaminate corn in the field, but can also infest wheat, barley, and oats [20]. The positive percentage and average concentration of the analyzed *Fusarium* mycotoxin FUM in the present study were rather high. In corn, the prevalence and average concentration of positives were 81.55% and 2618.81 μg/kg, respectively, which was similar to the results obtained in previous Chinese reports. In 2014, FB1 was detected as 100% positive with an average concentration of 116.5 μg/kg in the corn grains collected from the North China Plain [21]. In 2011, the FUMs’ positive rates in corn from Gansu, Shandong, Ningxia, and Inner Mongolia were 31.5%, 81.1%, 46.2%, and 53.6%, respectively, with concentrations ranging from ≤11 to 13,110 μg/kg [22], and in 2010, in the FUMs survey in corn from the main corn-producing provinces in China, FUM contamination exhibited the average concentrations of 3990, 845, and 665 μg/kg, respectively, in Liaoning, Shandong, and Henan [23].

Trichothecenes type B are represented by DON, which is produced by *F. graminearum*, *Gibberella zeae*, and *F. culmorum*, is the most common mycotoxin in cereals in China. These species can easily infest assorted cereal species, including corn, wheat, barley, and rice. Trichothecenes type B in this survey was highly detected in 70.56% of wheat and 95.31% of bran samples, with average concentrations of positives reaching 2129.29 μg/kg and 1469.40 μg/kg, respectively. With the representative studies on the mycotoxin contamination in wheat in China, DON was well proved to be most widespread, and even its contamination level varied with the average concentrations ranging from 240 μg/kg to 17,754 μg/kg depending on the different geographic regions of China [24,25]. DDGS exhibited the highest average levels of trichothecenes type B of all commodities, and this can be expected since the mycotoxins presented in the starting material can be extracted in DDGS. DON concentration by dry weight has been reported to be three times higher in DDGS than in the staring grains [26].

ZEN is mainly generated by *F. graminearum*, *F. culmorum*, *F. semitectum*, *F. equiseti*, and *F. cerealis*. ZEN is widely spread and easily contaminates barley, wheat, and soybeans, and especially corn, which is considered to be most susceptible to ZEN contamination. From 2017 to 2021, ZEN contamination was relatively severe in corn with a contamination level of 176.79 μg/kg, which was the highest value among all the investigated commodities. This result was supported by several published studies. From 2016 to 2017, the positive detection rate for ZEN in corn was 92.05% scaled in 21 provinces in China, and the positive average was 104.1 μg/kg in the year 2016 [18]. In another study that investigated the contamination of ZEN and its derivatives in corn, the positive rate for ZEN was 94% and the average concentration was 109.1 μg/kg [27]. Furthermore, the ZEN contamination level in grasses was relatively high, and even the prevalence was much lower than for other materials in this survey.

For all investigated mycotoxins, finished feeds exhibited high positive rates, especially for the *Fusarium* mycotoxins. Since the finished feeds are blends of different raw materials, it can be assumed that the finished feeds contain blends of mycotoxins which already exist in these starting materials. In addition, some mycotoxins can be produced during processing or storage. Thus, more studies are necessary to focus on the comprehensive survey on the patterns of mycotoxin contamination on different types of finished feeds and further assessment of feed safety and mycotoxin risk levels. In 2016, the positive rates for ZEN, AFB1, and DON were as high as 99.5%, 100%, and 100%, respectively [19], and it has also been reported that the detection rate of AFB1 in 200 dairy cow concentrates collected from 10 provinces in China reached 42% [28]. Compared with the previously obtained results, which showed a more severe mycotoxin exposure, the toxin prevalence was significantly reduced in the last 5 years, especially for aflatoxins. The knowledge and awareness about mycotoxin contamination is growing in China.

### 3.2. Regional Patterns of Mycotoxin Occurrence in New-Season Corn

Corn is one of the most important grains applied as a main feed raw materials in China. As presented in this survey, the prevalence and contamination levels of mycotoxins varied between years, and more significantly, regions also exhibited different patterns of mycotoxin occurrence. The large differences in mycotoxin occurrence could be due to the great diversity in climate conditions [29,30,31,32], together with multiple cultural practices across China, and as a large corn-producing country with complicated changing climatic conditions, different corn-producing arears could be favored by different toxic fugal growth and present different mycotoxin occurrence trends.

The regions of the Anhui, Jiangsu, and Henan provinces exhibited high prevalence and concentrations of aflatoxins, and the high frequency remained relatively stable from 2017 to 2021, which led to high exposure levels and a huge burden for animal production. It has been reported that crops are particularly susceptible to aflatoxin contamination during periods of drought, and the contamination can occur before harvest and continue to increase post-harvest under hot and humid conditions [32]. The variation of aflatoxin contamination among different regions reflects these associations well. The contamination level of aflatoxins was higher in the Anhui and Jiangsu provinces than in other investigated regions, and this could be largely due to the higher temperature and greater humidity in these semi-tropical southerner regions of China. In the global mycotoxin survey of a 10-year period, aflatoxins were detected prevalently at high concentrations in southern regions worldwide of Sub-Saharan Africa, Southeast Asia, and South Asia [11], which partly supported this result. Due to the climate changes, the contamination levels of aflatoxins presented regular variation trends from year to year. Generally, the concentrations spiked in the year 2018 in most of the investigated regions, and this could be associated with the more frequent typhoons during this period, coinciding with high rainfall in July and August leading to corn harvest in 2018. In the Henan province of Central China, the aflatoxin contamination level in 2017 was high, and this could be due to the relatively high temperature and high precipitation observed in August and September in 2017 which happened to be the harvest and storage periods of corn in that region.

The comparison of FUM occurrence showed that Shandong, Anhui, and Henan provinces obtained the highest contamination levels. Since FUM contamination can be favored by high temperatures and low precipitation around silking period [33], the year-by-year variation of FUM concentrations can be expected with the regular changes of weather conditions. Interestingly, the year-by-year variation trends of FUM concentrations exhibited the similar trends of aflatoxins in this survey, which suggested the similar key impacts of high temperature and drought during corn-producing periods on the levels of both aflatoxin and FUM contamination. Particularly in Hebei province of North China, a peak in average concentrations of FUM observed in 2018 could be associated with a higher temperature observed in summer, and at the same time Hebei happened to suffer from less precipitation, which may facilitate FUM production.

The occurrence of trichothecenes type B is high when corn flowers and matures with mild temperature and persistent rainfall, with high relative humidity being most important [32,34]. Accordingly, higher concentrations of trichothecenes type B were detected in samples from the temperate regions globally of North America, Northern Europe, Central Europe, and East Asia [11]. When the survey scope narrowed down to the main corn-producing areas in China, the northerner regions also presented a higher occurrence [1,15]. Trichothecenes type B concentrations in new-season corn varied from year to year in several regions. In Shandong province of East China, the concentration was exceptionally high in corn harvested in 2021. This peak value could correspond to the heavy rainfall caused by the typhoon attack in July 2021 during the main silking period of corn, followed by persistent relatively high precipitation in August, which is generally the main corn harvest and storage period. As recorded, the average precipitation in Shandong province was 778.9 mm from June to October in 2021, 64.5% more than that in the same period, and same as in Hebei province. The average precipitation of Hebei province in July reached 295.6mm, more than twice the annual average, which strengthened the key impact of high precipitation on the fugal growth and trichothecenes type B production.

ZEN is also mainly produced by the same DON-generating species and widely spread in temperate regions. For ZEN, the favored conditions are humidity and mild temperatures [11]. Similar with the trichothecenes type B contamination, ZEN contamination was more severe in northerner regions than others, where Shandong province was found to have the highest prevalence and concentrations, with the highest level up to 722.00 µg/kg and a positive rate of 88.00% in the year 2021.

### 3.3. Co-Occurrence of Mycotoxins in New-Season Corn

Generally, feeds and materials can be contaminated by different varieties of mycotoxins. With the development of mycotoxin research, there is increasing awareness of the need to take the co-occurrence of multi-mycotoxins into consideration. The co-exposure of mycotoxins could lead to additive or synergistic health risks as previously reported [35,36,37].

The co-occurrence of *Fusarium* mycotoxins was very common. In this survey, the most frequently observed mycotoxin combinations were *Fusarium* mycotoxins, especially trichothecenes type B and ZEN. Previously published studies suggested the high degrees of co-occurrence, e.g., on a global basis, the co-contamination ratios of DON and ZEN were 48%, 39%, and 28%, respectively, in samples of finished feeds, corn, and maize collected from 2008 to 2017 [11], while in China, the co-contamination ratio was reported to be 94.0% in all feed samples in 2011 [1]. We also calculated the correlation of concentrations for the combination of any two mycotoxins in new-season corn. The concentrations of trichothecenes type B and ZEN exhibited a highly positive correlation in the last five years. Even in a mycotoxin survey conducted in Africa, the concentrations of DON and ZEN were also detected with positive correlation coefficients of 0.543 and 0.319, respectively, in two different regions of Africa [14]. Therefore, co-contamination to animals is the rule rather than the exception on a global basis, and it is important to take into account the combined toxic effects of mycotoxins [11].

In addition, with the advances in multi-mycotoxin detection methods, there are also masked mycotoxins (e.g., 3- and 15-acetylated DON and DON glucoside) and new emerging mycotoxins (e.g., enniatins and beauvericin) detected in feeds and materials [12]. It has been reported that DON is usually well correlated with DON-3-glucoside and so is ZEN with ZEN-14-sulfate. Therefore, more studies are need to focus on detecting the derivatives of DON and ZEN, and their co-contamination and correlations should be further investigated in future studies.

## 4. Conclusions

In conclusion, the mycotoxin survey of 9329 samples in China from 2017 to 2021 indicated that mycotoxins are ubiquitously present in feeds. Each feed exhibited a distinct pattern of mycotoxin occurrence. Corn was detected with high contamination levels of *Fusarium* mycotoxins, wheat and bran were typically affected by trichothecenes type B, peanut meals were highly susceptible to aflatoxins, and corn by-products were concentrated with trichothecenes type B and ZEN. What should be paid special attention to is the fact that, as blends of feed raw materials, finished feeds exhibited a comparatively high prevalence of all mycotoxins.

New-season corn was commonly infested by mycotoxins and presented varied patterns of mycotoxin occurrence in different corn-producing regions of China. Climatic conditions are regarded as one of the determinant factors and drive distinct patterns of mycotoxin occurrence in each region. As reported in this survey, in Shandong province of East China, large fractions of new-season corn were polluted with high concentrations of *Fusarium* mycotoxins, and in the Anhui and Henan provinces, there were high contamination risks detected in aflatoxins and FUMs. From year to year, the levels of mycotoxin contamination varied along with the changes of regular climate conditions. For example, in southern regions of China, unusual high temperatures and/or heavy rainfalls caused by typhoon attacks could cause aflatoxins levels far in excess of the concentrations typically observed, and if hot weather and drought occur in summer, there could be a relatively high level of FUM pollution in corn. In temperate regions, if persistent rainfall occurs during the corn flowering and ripening periods, more severe exposures of DON and ZEN should be expected in that year. Mathematical modelling has been established and showed the potential changes in aflatoxin occurrence in Europe [38], while in China the modelling approaches and data for mycotoxin prediction are limited. With this perspective, the priority is to obtain solid scientific evidence on correlations of climate change and mycotoxin occurrence to allow for a comprehensive and appropriate mycotoxin risk warning system.

The co-occurrence of multi-mycotoxins generally exists in feeds. The risk assessment of co-exposure of mycotoxins is a significant challenge for the future. Studies have revealed that mycotoxins, not only regulated mycotoxins, but masked and new emerging toxins in feeds, potentially aggravate or change the negative effects on animal health [12]. Therefore, advanced detection tools, standardized analytical methods, and scientific research on health risks of combined mycotoxins are urgently required. In addition, strengthened mycotoxin surveillance and effective measures of mycotoxin prevention and control are vital to ensure feed and food safety in China. The application of targeted detoxification products is considered to be one of the most efficient, safe, and environmentally friendly measures for mycotoxin reduction. Meanwhile, good agriculture practices in all sectors, a mycotoxin early warning system, and further, a comprehensive mycotoxin control system should act together to ensure effective mycotoxin management and reduced public health risks.

## 5. Materials and Methods

### 5.1. Collection of Samples

A total of 9392 samples of feeds and raw materials were collected from 2017 to 2021 in a 5-year period. The dataset comprised 1188, 2084, 1867, 1610, and 2643 samples in each year from 2017 to 2021, respectively, and included samples of finished feed, such as poultry and swine feeds, samples of feed raw material, such as corn and wheat, samples of corn by-product, such as DDGS, and samples of grasses, such as silage and alfalfa. Samples of new-season corn were included in the corn samples and the dataset comprised 1160 samples collected from 10 provinces of Heilongjiang, Jilin, Liaoning provinces in Northeast China; Inner Mongolia and Hebei provinces in North China; Shandong, Jiangsu, and Anhui provinces in East China; Henan in Central China; and Gansu province in Northwest China. Sample numbers per commodity and new-season corn samples per region are given in Table 6 and Table 7, respectively.

Complete notes with detailed information surrounding the circumstances of the samples, including temperature, moisture, and water content were submitted with the samples. Original samples of 1 kg were collected and kept at 4 °C before being transported to the analytical Romer Labs in Wuxi, China [1]. Sampling, milling, and homogenization of a 100 g representative sub-sample were performed as described previously [12].

### 5.2. Mycotoxin Analysis

Samples from year 2017 with relatively complex matrices (n = 1012), such as finished feeds, were analyzed by Romer Labs in Wuxi, China. High Performance Liquid Chromatography (HPLC) was applied for the determination of mycotoxin concentrations of aflatoxins, trichothecenes type B, FUMs, and ZEN. Sample preparation, chromatographic conditions and parameters, and system program were specified by Guan et al. [1]. Limits of detection (LODs) for each mycotoxin were 0.3 μg/kg for AFB1, 0.1 μg/kg for AFB2, 0.1 μg/kg for AFG1, 0.1 μg/kg for AFG2, 10 μg/kg for ZEN, 50 μg/kg for DON, 25 μg/kg for FB1, and 25 μg/kg for FB2.

From the year 2018 to the year 2021, a total of 5996 samples with relatively complex matrices were determined with liquid chromatography—tandem mass spectrometry (LC-MS/MS) for the analysis of mycotoxin concentrations. LC-MS/MS is a multi-mycotoxin analysis method and a total of 18 mycotoxins, including 4 kinds of aflatoxins (AFB1, AFB2, AFG1, and AFG2), ZEN, 5 kinds of trichothecenes type B (DON, 3-Acetyl-Deoxynivalenol, 15-Acetyl-Deoxynivalenol, Nivalenol, and Fusarenon X), 4 kinds of trichothecenes type A (T-2, HT-2 toxin, Diacetoxyscirpenol, and Neosolaniol), 3 kinds of fumonisins (FB1, FB2, and FB3), and OTA can be detected simultaneously [12]. Procedures of analysis and quality control were performed as described previously [15]. LODs of this method for each mycotoxin were 0.5 μg/kg for AFB1, 0.5 μg/kg for AFB2, 0.5 μg/kg for AFG1, 0.5 μg/kg for AFG2, 10 μg/kg for ZEN, 10 μg/kg for DON, 10 μg/kg for FB1, 10 μg/kg for FB2, and 10 μg/kg for FB3.

For the remaining 2384 samples of feed raw materials with relatively simple matrices, such as corn, wheat, and bran, the method of enzyme-linked immunosorbent assay (ELISA) was performed for the analysis of mycotoxin concentrations. Procedures of sample preparation and analysis were operated with a commercially available test kit (AgraQuant^®^ Assay, Romer Labs Diagnostic GmbH, Tulln, Austria) according to its operating instructions. LODs of this method for each mycotoxin were 2 μg/kg for AFB1, 25 μg/kg for ZEN, 250 μg/kg for DON, and 250 μg/kg for FUMs.

For all analyzed samples, when the mycotoxin concentrations were detected with higher values than the following threshold, the samples can be defined as mycotoxin-positive. The concentration threshold for each mycotoxin was defined as 1 μg/kg for the sum of AFB1, AFB2, AFG1, and AFG2, 32 μg/kg for ZEN; 50 μg/kg for trichothecenes type B; and 100 μg/kg for the sum of FB1, FB2, and FB3.

### 5.3. Statistical Analysis

Statistical analyses were performed using JMP 10.0 Statistics software (SAS Institute, Cary, NC, USA). Raw data were verified for normality and homogeneity of variance, and non-Gaussian distribution of the mycotoxin concentrations was presented in this survey. Therefore, the non-parametric test (Wilcoxon test) was applied for the variance analysis of raw data. Different letters in the same column indicate significant differences in the average of positives. Significance was declared at *p* ≤ 0.05 for all variables.

Analysis for year-to-year variations of the positive rates and average concentrations for each mycotoxin were performed for the feed commodities or regions with a sufficiently high number of samples, for the purpose of preventing single samples from exerting an unduly strong influence on the overall results.

Correlations between any two mycotoxin concentrations were analyzed with the ggpairs in the ggally package using R software, version 3.3.0 [39,40]. In the correlation analysis, values below the LODs were considered as zero values in this study.

## Figures and Tables

**Figure 1 toxins-15-00063-f001:**
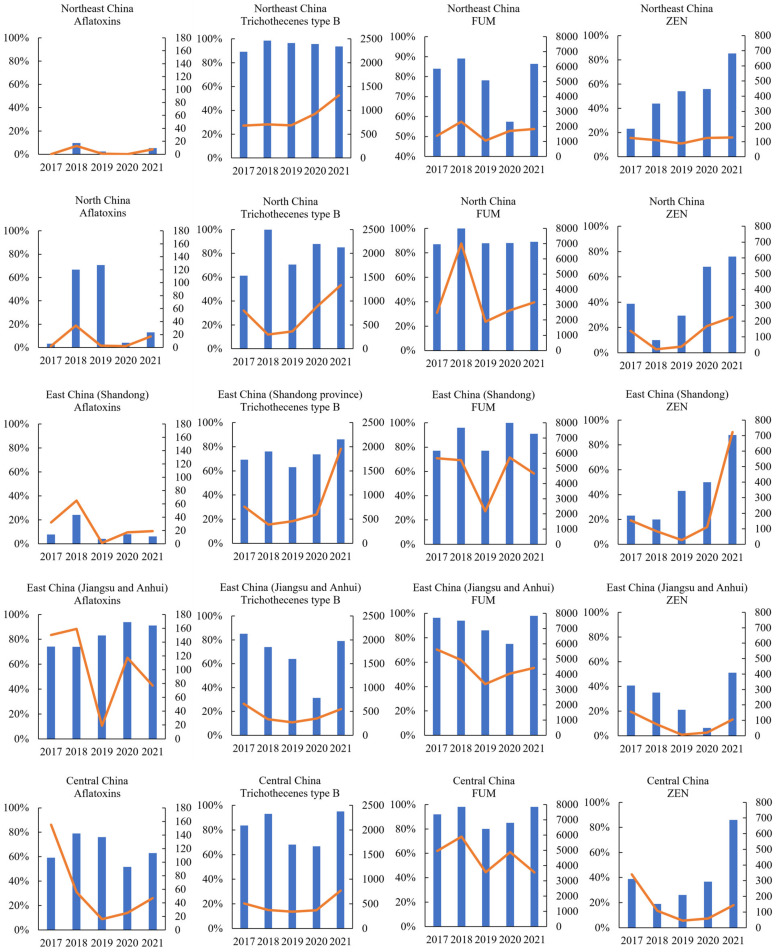
Year-to-year variation of mycotoxin concentrations in new-season corn from different regions of China. The horizontal axis showed harvest years from 2017 to 2021. Vertical axes exhibited percentages of positive rates with column charts and values of positive averages with line charts. Data points are shown if ≥20 samples per region were available.

**Figure 2 toxins-15-00063-f002:**
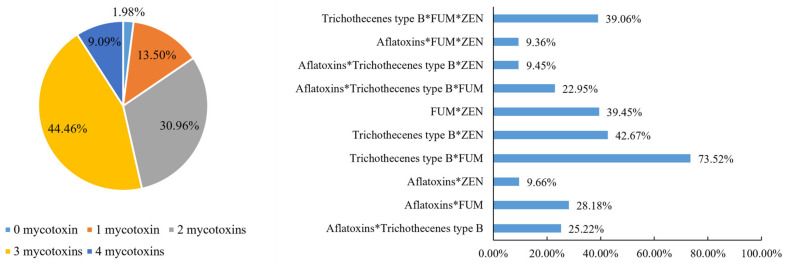
Mycotoxin co-occurrence in new-season corn from 2017 to 2021. Pie chart presented the percentages of different numbers of mycotoxin co-occurrence in new season corn samples from 2017 to 2021; Bar chart presented the percentages of different combinations of two to three mycotoxins in new season corn samples from 2017 to 2021. *, combination of different mycotoxins.

**Figure 3 toxins-15-00063-f003:**
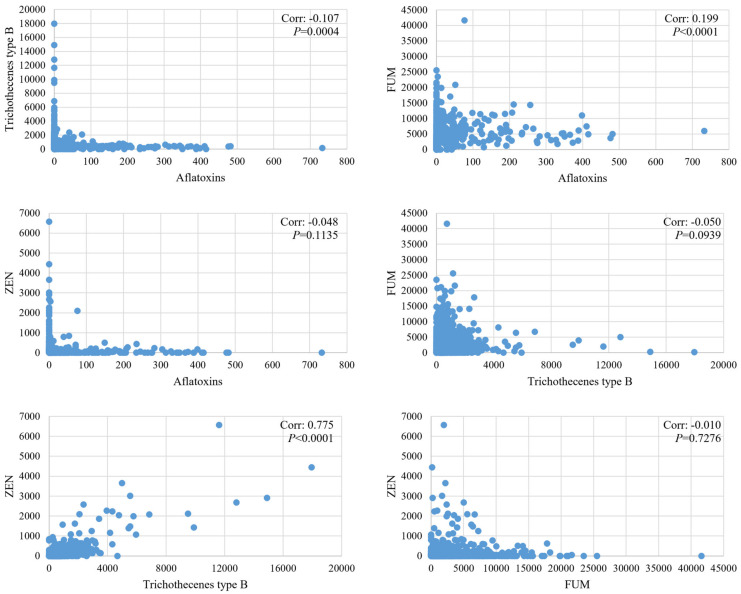
Correlation of mycotoxin concentrations in new-season corn from 2017 to 2021.

**Table 1 toxins-15-00063-t001:** Mycotoxin legislation in China (hygienical standard for feeds GB13078–2017).

Item	Matrix		Allowable Level
Aflatoxin B1, μg/kg	Feed material	Corn processing products, peanut cake (meal)	≤50
		Vegetable oil (excluding corn oil and peanut oil)	≤10
		Corn oil, peanut oil	≤20
		Other vegetable feed materials	≤30
	Feed product	Concentrated feeds for piglets and young birds	≤10
		Concentrated feeds for meat ducklings in later stage, growing ducks and laying ducks	≤15
		Other concentrated feeds	≤20
		Concentrated supplements for calves and lambs	≤20
		Concentrated supplements during lactation	≤10
		Other concentrated supplements	≤30
		Compound feeds for piglets and young birds	≤10
		Compound feeds for meat ducklings in later stage, growing ducks, and laying ducks	≤15
		Other formulated feeds	≤20
OTA, μg/kg	Feed material	Cereals and processed products	≤100
	Feed product	Compound feeds	≤100
Zearalenone, μg/kg	Feed material	Corn and its processed products (excluding corn husk, sprayed corn husk, and corn starch powder)	≤500
		Corn husk, sprayed corn husk, corn starch powder, and corn distiller’s grains	≤1500
		Other vegetable feed materials	≤1000
	Feed product	Concentrated supplements for calves, lambs, and lactation	≤500
		Compound feeds for piglets	≤150
		Compound feeds for young sows	≤100
		Compound feeds for other pigs	≤250
Deoxynivalenol, μg/kg	Feed material	Vegetable feed materials	≤5000
	Feed product	Concentrated supplements for calves, lambs, and lactation	≤1000
		Other concentrated supplements	≤3000
		Compound feeds for pigs	≤1000
		Other compound feeds	≤3000
T-2 toxin, μg/kg	Feed material	Vegetable feed materials	≤500
	Feed product	Compound feeds for pigs and poultry	≤500
Fumonisins (B1 + B2), μg/kg	Feed material	Corn and its processed products, corn distiller’s grains products, corn silage, and corn straw	≤60
	Feed product	Concentrated supplements for calves and lambs	≤20
		Concentrated supplements for horses and rabbits	≤5
		Other ruminant concentrated supplements	≤50
		Concentrated feeds for pigs	≤5
		Concentrated feeds for poultry	≤20
		Compound feeds for pigs, rabbits, and horses	≤5
		Compound feeds for poultry	≤20
		Compound feeds for fish	≤10

**Table 2 toxins-15-00063-t002:** General mycotoxin contamination from 2017 to 2021.

	Aflatoxins	Trichothecenes Type B	FUMs	ZEN
2017–2021				
Sample number	9329	9329	9057	9329
Positive rate	17.44%	87.07%	77.14%	56.29%
Average of positives (μg/kg)	49.80	838.89	1842.57	183.81
Maximum (μg/kg)	10,091	59,325	60,276	11,245
Year 2017				
Sample number	1188	1188	916	1188
Positive rate	12.71%	98.74%	71.07%	33.42%
Average of positives (μg/kg)	87.15 ^a^	748.21 ^b^	1660.76 ^b^	300.57 ^a^
Maximum (μg/kg)	476	13,206	16,976	5416
Year 2018				
Sample number	2083	2083	2083	2083
Positive rate	25.25%	88.72%	80.60%	46.86%
Average of positives (μg/kg)	81.13 ^a^	843.18 ^d^	2464.74 ^a^	209.19 ^b^
Maximum (μg/kg)	10,091	59,325	60,276	8247
Year 2019				
Sample number	1828	1828	1828	1828
Positive rate	20.46%	88.24%	84.57%	57.44%
Average of positives (μg/kg)	26.79 ^b^	614.10 ^c^	1816.52 ^b^	74.20 ^c^
Maximum (μg/kg)	497	12,907	56,332	4686
Year 2020				
Sample number	1604	1604	1604	1604
Positive rate	15.52%	72.44%	62.91%	46.82%
Average of positives (μg/kg)	33.59 ^b^	642.52 ^b^	1562.75 ^b^	204.84 ^b^
Maximum (μg/kg)	482	10,426	30,872	11,245
Year 2021				
Sample number	2626	2626	2626	2626
Positive rate	12.45%	88.61%	80.05%	79.09%
Average of positives (μg/kg)	20.82 ^b^	1135.07 ^a^	1555.39 ^b^	197.37 ^b^
Maximum (μg/kg)	440	13,513	38,563	10,467
Source of variance (year)				
*p*-values (Average of positives)	<0.01	<0.01	<0.01	<0.01

^a–d^ in the average of positives of the same column showed significantly different (*p* < 0.05).

**Table 3 toxins-15-00063-t003:** Mycotoxin contamination in different feed commodities from 2017 to 2021.

		Aflatoxins	Trichothecenes Type B	FUMs	ZEN
Raw materials					
Corn	Sample number	2873	2873	2873	2873
	Positive rate	16.50%	87.47%	81.55%	51.51%
	Average of positives (μg/kg)	63.28	871.28	2618.81	176.79
	Maximum (μg/kg)	733	12,808	40,090	4686
Wheat	Sample number	411	411	411	411
	Positive rate	1.22%	70.56%	21.41%	48.18%
	Average of positives (μg/kg)	2.60	2129.29	332.31	105.50
	Maximum (μg/kg)	5	59,325	910	1205
Soybean meal	Sample number	257	257	257	257
	Positive rate	7.78%	14.79%	6.61%	39.69%
	Average of positives (μg/kg)	4.65	171.50	760.24	45.26
	Maximum (μg/kg)	35	597	6932	237
Peanut meal	Sample number	69	69	69	69
	Positive rate	100.00%	4.35%	7.25%	4.35%
	Average of positives (μg/kg)	417.72	77.67	50.40	37.33
	Maximum (μg/kg)	10,091	139	120	61
Bran	Sample number	341	341	341	341
	Positive rate	4.11%	95.31%	6.74%	23.75%
	Average of positives (μg/kg)	2.64	1469.40	214.48	78.33
	Maximum (μg/kg)	4	13,206	1069	619
Rice bran meal	Sample number	64	64	64	64
	Positive rate	15.63%	89.06%	10.94%	51.56%
	Average of positives (μg/kg)	67.40	307.93	341.29	106.73
	Maximum (μg/kg)	214	1458	583	847
Cottonseed meal	Sample number	104	104	104	104
	Positive rate	35.58%	6.73%	43.27%	7.69%
	Average of positives (μg/kg)	40.92	127.71	318.69	13.50
	Maximum (μg/kg)	531	557	3812	200
Corn by-products					
DDGS	Sample number	197	197	197	197
	Positive rate	26.40%	98.48%	86.80%	87.82%
	Average of positives (μg/kg)	31.94	2327.08	6065.33	560.16
	Maximum (μg/kg)	179	12,907	59,642	5416
Corn gluten meal	Sample number	78	78	78	78
	Positive rate	35.90%	84.62%	100.00%	94.87%
	Average of positives (μg/kg)	75.36	581.73	8363.50	2004.30
	Maximum (μg/kg)	469	2619	60,276	11,245
Grasses					
Silage	Sample number	678	678	582	678
	Positive rate	0.44%	87.02%	83.16%	61.80%
	Average of positives (μg/kg)	8.67	1207.96	716.88	208.55
	Maximum (μg/kg)	21	18,273	6533	8649
Alfalfa	Sample number	161	161	105	161
	Positive rate	4.97%	55.28%	36.19%	19.88%
	Average of positives (μg/kg)	115.00	317.11	163.18	339.69
	Maximum (μg/kg)	291	1049	281	2852
Oat grass	Sample number	124	124	106	124
	Positive rate	0.00%	53.23%	38.68%	36.29%
	Average of positives (μg/kg)	-	1728.85	381.76	484.53
	Maximum (μg/kg)	0	9363	1986	2622
Finished feeds					
Poultry feed	Sample number	1857	1857	1857	1857
	Positive rate	29.99%	98.82%	95.85%	71.46%
	Average of positives (μg/kg)	15.79	563.88	1437.82	120.13
	Maximum (μg/kg)	206	6176	17,052	1490
Swine feed	Sample number	1418	1418	1418	1418
	Positive rate	21.93%	97.53%	92.95%	57.12%
	Average of positives (μg/kg)	16.95	513.82	1116.00	83.39
	Maximum (μg/kg)	245	3620	13,254	857
Concentrate supplement	Sample number	207	207	138	207
	Positive rate	7.73%	97.10%	87.68%	53.14%
	Average of positives (μg/kg)	3.13	618.31	965.70	93.52
	Maximum (μg/kg)	14	5258	7669	717
TMR	Sample number	470	470	437	470
	Positive rate	2.77%	95.74%	93.36%	71.91%
	Average of positives (μg/kg)	8.54	621.77	554.15	89.36
	Maximum (μg/kg)	39	6492	7619	719

**Table 4 toxins-15-00063-t004:** Mycotoxin contamination in new-season corn from 2017 to 2021.

	Aflatoxins	Trichothecenes Type B	FUMs	ZEN
2017–2021				
Sample number	1153	1153	1153	1153
Positive rate	31.40%	84.04%	87.16%	50.13%
Average of positives (μg/kg)	63.83	807.12	3549.65	203.41
Maximum (μg/kg)	733	6849	41,600	3017
Year 2017				
Sample number	176	176	176	176
Positive rate	28.98%	80.68%	88.07%	32.95%
Average of positives (μg/kg)	147.74 ^a^	650.35 ^b^	3602.13 ^bc^	205.28 ^b^
Maximum (μg/kg)	476	3010	41,600	2095
Year 2018				
Sample number	216	216	216	216
Positive rate	46.76%	91.20%	94.44%	28.70%
Average of positives (μg/kg)	72.67 ^b^	483.56 ^b^	4743.70^a^	97.28 ^b^
Maximum (μg/kg)	733	2530	20,837	454
Year 2019				
Sample number	207	207	207	207
Positive rate	32.37%	78.74%	80.68%	40.58%
Average of positives (μg/kg)	10.46 ^c^	532.03 ^b^	1944.99 ^c^	62.88 ^b^
Maximum (μg/kg)	118	2610	17,040	582
Year 2020				
Sample number	207	207	207	207
Positive rate	24.15%	77.29%	78.26%	46.86%
Average of positives (μg/kg)	51.96 ^b^	706.16 ^b^	3941.64 ^b^	113.91 ^b^
Maximum (μg/kg)	482	4670	23,480	1572
Year 2021				
Sample number	347	347	347	347
Positive rate	26.80%	88.47%	91.35%	79.83%
Average of positives (μg/kg)	53.05 ^b^	1285.93 ^a^	3400.62 ^c^	300.73 ^a^
Maximum (μg/kg)	331	6849	18,320	3017
Source of variance (year)				
*p*-values (average of positives)	<0.01	<0.01	<0.01	<0.01

^a–c^ in the average of positives of the same column showed significantly different (*p* < 0.05).

**Table 5 toxins-15-00063-t005:** Mycotoxin contamination in new-season corn of different regions from 2017 to 2021.

	Aflatoxins	Trichothecenes Type B	FUMs	ZEN
Northeast China				
Sample number	379	379	379	379
Positive rate	3.69%	94.99%	79.42%	55.67%
Average of positives (μg/kg)	9.36 ^c^	889.63 ^b^	1678.28 ^c^	115.44 ^b^
Maximum (μg/kg)	44	4670	11,193	726
North China				
Sample number	199	199	199	199
Positive rate	30.15%	80.90%	89.95%	49.75%
Average of positives (μg/kg)	15.36 ^c^	842.73 ^b^	3379.15 ^b^	176.69 ^b^
Maximum (μg/kg)	256.63	3430	18,320	1572
East China (Shandong)				
Sample number	191	191	191	191
Positive rate	8.38%	76.96%	90.05%	59.16%
Average of positives (μg/kg)	35.51 ^c^	1207.20 ^a^	4699.77 ^a^	483.87 ^a^
Maximum (μg/kg)	188.00	6849.00	22,200.00	3017.00
East China (Anhui and Jiangsu)				
Sample number	138	138	138	138
Positive rate	82.61%	71.01%	92.03%	35.51%
Average of positives (μg/kg)	103.08 ^a^	479.16 ^c^	4606.60 ^a^	100.24 ^b^
Maximum (μg/kg)	733.00	2310.00	19,800.00	563.87
Central China				
Sample number	246	246	246	246
Positive rate	64.23%	82.52%	91.87%	43.09%
Average of positives (μg/kg)	61.62 ^b^	501.16 ^c^	4707.84 ^a^	152.18 ^b^
Maximum (μg/kg)	605.00	2430.00	41,600.00	2095.36
Source of variance (region)				
*p*-values (Average of positives)	<0.01	<0.01	<0.01	<0.01

^a–c.^ in the average of positives of the same column showed significantly different (*p* < 0.05).

**Table 6 toxins-15-00063-t006:** Total sample numbers per commodity per year.

	Year 2017	Year 2018	Year 2019	Year 2020	Year 2021	Total
Raw materials						
Corn	289	540	469	665	910	2873
Wheat	-	113	-	72	226	411
Soybean meal	23	66	49	63	56	257
Peanut meal	-	36	-	25	8	69
Bran	59	102	66	49	65	341
Rice bran meal	31	-	12	9	12	64
Cottonseed gluten meal	-	40	51	-	13	104
Flaked Corn	-	21	-	-	-	21
Corn by-products						
DDGS	65	50	40	15	27	197
Corn gluten meal	-	20	22	27	20	89
Sprayed corn husk	-	-	7	3	11	21
Corn germ meal	-	-	10	-	-	10
Grasses						
Silage	96	118	116	105	243	678
Alfalfa	56	36	14	39	16	161
Oat	18	-	12	30	64	124
Soybean husk	-	-	11	3	6	20
Finished feeds						
Poultry feed	155	353	645	270	434	1857
Swine feed	294	468	182	124	350	1418
Concentrate supplement	69	38	40	26	34	207
Total mixed ration (TMR)	33	83	121	85	148	470

**Table 7 toxins-15-00063-t007:** Samples of new-season corn per region per year.

	Year 2017	Year 2018	Year 2019	Year 2020	Year 2021	Total
Heilongjiang	12	11	12	21	27	83
Jilin	16	21	17	11	28	93
Liaoning	28	41	58	36	40	203
Hebei	19	30	29	25	72	175
Shandong	13	25	34	38	81	191
Henan	49	57	24	60	56	246
Anhui and Jiangsu	27	31	21	16	43	138
Inner Mongolia	12	-	12	-	-	24
Gansu	-	-	-	7	-	7

## Data Availability

Not applicable.

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
