# Peer review of "Mycotoxin Occurrence in Feeds and Raw Materials in China: A Five-Year Investigation"

_toxins, 2023, doi:10.3390/toxins15010063_

Round 1
Reviewer 1 Report
The article contains some quite important points that need to be improved, as follows.
Defining the study as a "survey" is not appropriate.
No real statistical analyses were carried out (only correlations without a P value).
The statistical analyses should be made and should take into account the following items:
- the different methods of analysis
- the time
- the influence of truncate values of outcomes
- the missing data (their type should be indentified and a suitable method should be performed to manage them)
- the effect of grouping of the small samples collected across various categories (regions etc.)
- the randomization and sample collection methods
- the probably non-Gaussian distribution of concentrations (log-normal?)
Moreover:
What is the strength of the article compared to previous information on mycotoxins?
According to my opinion Authors should take into account the points described here above.
Author Response
Point 1: The article contains some quite important points that need to be improved, as follows.
Defining the study as a "survey" is not appropriate.
No real statistical analyses were carried out (only correlations without a P value).
The statistical analyses should be made and should take into account the following items:
- the different methods of analysis
- the time
- the influence of truncate values of outcomes
- the missing data (their type should be indentified and a suitable method should be performed to manage them)
- the effect of grouping of the small samples collected across various categories (regions etc.)
- the randomization and sample collection methods
- the probably non-Gaussian distribution of concentrations (log-normal?)
Response 1: Thank you for your suggestions. As you suggested, statistical analysis will make the data more concise and convincing, and the differences of mycotoxin contamination levels in different regions and different years will be more obviously presented through statistical analysis. However, considering that the main purpose of this article is to improve the public awareness of the feed mycotoxin contamination in China, a relatively simple data processing method was applied in this article to present more detailed data, which might be more intuitive to the readers. As for the different methods of analysis, due to the relatively large time span of the survey, different methods with different LODs are applied to analyze the mycotoxin contamination. To eliminate the difference, a truncate value was set for each mycotoxin, and a sample will be judged as positive if the determined concentration exceeds the truncate value. As for the missing data, in order to present the information as detailed as possible, the data from some regions or feed raw materials with relatively small sampling numbers are eliminated, which has little impact on the overall data analysis in this paper. We also applied standard sampling methods, as described in the relevant references, to minimize the differences caused by sampling errors. I hope this respond could meet your requirements. Of course, this is just my personal views. If I don't understand your suggestions accurately, please let me know and I will make corresponding modifications as required.
Point 2: What is the strength of the article compared to previous information on mycotoxins?
Respond 2: As for the strength of this article, we believe that although lots of mycotoxin surveys have been carried out in various regions on a scale of global, including China, there are still limited articles published. This article obtained the largest dataset for now in China, including mycotoxin pollution of various feed raw materials and finished products in recent five years. We also analyzed the dynamic changes of mycotoxin pollution levels in different regions of new season corn in different years, and the results may have important implications for the future trends of mycotoxin contamination in China Through detailed data presentation in this article, we aim for the public to improve the awareness of mycotoxin contamination, thus, to adopt the effective mycotoxin management timely and reduce public health risk and the resulting economic losses.

Reviewer 2 Report
Dear Authors,
The manuscript entitled “Mycotoxin occurrence in feeds and raw materials in China: A five-year investigation” presents timely and adequate information that is relevant to the readers. The authors have explained that mycotoxins contamination of feeds are common, whose occurence may be related with climate and weather conditions.
The introduction is well-written and organized. The overall manuscrit is also well presented and supported by relevant references. I enjoyed reading it, as it is easly understandable and appealling for the readers.
I have some minor corrections:
Line 10: Fusarium in italics
Line 27: Please place keywords alphabetically
Line 97: Fusarium in italics
Line 491: Never start a sentence with a number. Rewrite the sentence to something like this: Samples from year 2017 with relatively complex matrices (n=1012), ...
Line 540: Ambrosia in italics
I also recommend the authors to revise the reference list, and check all the issue numbers of the Journals. It should be in italics.
Kind Regards
Author Response
Point 1: Line 10: Fusarium in italics
Respond 1: Fixed. Font has been changed to italics. Please see Line 10.
Point 2: Line 27: Please place keywords alphabetically
Respond 2: Keywords have been sorted according to their criticality and frequency in this manuscript. No changes have been made, please let me know if it’s a request. Thank you.
Point 3: Line 97: Fusarium in italics
Respond 3: Fixed. Font has been changed to italics. Please see Line 101.
Point 4: Line 491: Never start a sentence with a number. Rewrite the sentence to something like this: Samples from year 2017 with relatively complex matrices (n=1012),
Respond 4: The sentence has been changed to “Samples from year 2017 with relatively complex matrices (n=1012)”. Please see Line 495.
Point 5: Line 540: Ambrosia in italics
Respond 5: Fixed. Font has been changed to italics. Please see Line 544.
Point 6: Reference: Revise the reference list.
Respond 6: the References list has been revised accordingly, please see the revised References part: Line 532-630.

Reviewer 3 Report
The authors present a manuscript which details the levels of various mycotoxins from a large study in China.
Overall the paper is well written with a good level of detail. I think the work could be published but there are a few points to consider
The abstract is quite well written with a lot of qualitative information. It may be good to add something about the range of levels detected and how these varied.
The Introduction - it would be good to have the levels of mycotoxins legislated for in the EU and China, rather than just mentioning that the levels are controlled.
This sentence in the introduction (line 66) is not clear "While the related fields were relatively less investigated in and available data is still scarce and insufficient."
It is an interesting argument that mycotoxins can be under the limit but combinations may still exert an effect - is there any published data to back this up? Also the authors mention a synergistic effect of multiple mycotoxins produced at sub-threshold levels. Again are there any references that have proposed or studied this effect?
The auhtors mention climatic conditions and the effects on mycotoxins again it would be good to back these up with references from previous studies even though it is generally accepted that this is a driver of production
Results section - In table 1 would it be more useful to give the range rather than just the maximum value?
Table 2 - same comment as above - also there is a typo in the table description - "materials"
Also in the table would it be a good idea to indicate what the positive threshold for mycotoxins are in China, so it is easy to assess the level of contamination. When the authors say positive do they mean over this threshold or do they simply mean detected at any level above the limit of detection of the technique?
Just a general question but why do the authors feel that Aflatoxins are much lower incidence than the other mycotoxins - is this just due to the difference in levels making low level Aflatoxin contamination difficult to detect?
For Figure 1 the geographic location and the discussion of the results is it possible to link these differences to weather etc.
Discussion - "The large differences in mycotoxin occurrence could be contrib-uted by the great diversity in climate conditions," This does not seem very certain are there studies that have linked climatic change to development of mycotoxins?
In the discussion or results it may be good to give the climatic conditions/rainfall/temp etc. prior to and during harvest. At the moment the discussion is quite general
Author Response
Point 1: Abstract: It may be good to add something about the range of levels detected and how these varied.
Response 1: The specific values of the levels of mycotoxins contamination have been added in the abstract part. Please see Line 16-20.
Point 2: The Introduction: it would be good to have the levels of mycotoxins legislated for in the EU and China, rather than just mentioning that the levels are controlled.
Respond 2: The limits for individual mycotoxins of different animal species legislated in China have been added in revised Table 1. This will facilitate to compare the determined mycotoxin concentrations with the regulated levels to see the whole picture of the mycotoxin contamination levels in China. The specific values of mycotoxin limits in EU legislation can be referred to the References. We have not presented the mycotoxin levels in EU, because except for aflatoxins, the levels of other mycotoxins are recommended in EU, and also the legislated levels in China also refer to the EU guidance levels. Considering that this survey was mainly scaled in China, we thought it might be more focused if we just showed the legislated mycotoxin levels in China. Please let me know if we should add the EU mycotoxin guidance levels, and I will revise that later. The addition of revised Table 1 please see Line 93 and revised Table 1. The original table numbers are also changed accordingly.
Point 3: This sentence in the introduction (line 66) is not clear "While the related fields were relatively less investigated in and available data is still scarce and insufficient."
Respond 3: The sentence has been changed to “While the available data is still scarce and insufficient”. Please see Line 68-69.
Point 4: It is an interesting argument that mycotoxins can be under the limit but combinations may still exert an effect - is there any published data to back this up? Also the authors mention a synergistic effect of multiple mycotoxins produced at sub-threshold levels. Again are there any references that have proposed or studied this effect?
Respond 4: The references have been supplemented in the text and the Reference list. Please see Line 68, Reference 8,9,11, and Line 411, Reference 35-37.
Point 5: The authors mention climatic conditions and the effects on mycotoxins again it would be good to back these up with references from previous studies even though it is generally accepted that this is a driver of production
Respond 5: The supporting references of Aflatoxins, FUM and DON favoring weather conditions have been already described in the text, please see Line 355 (Reference 32),Line 375 (Reference 33), Line386 (Reference 32, 34). The references about ZEN have also been supplemented in the text, please see LINE 403.
Point 6: Results section - In table 1 would it be more useful to give the range rather than just the maximum value?
Respond 6: Thank you very much for your suggestion. It would be much more intuitive if we list the range of the mycotoxin concentrations. While, actually in all classes of the mycotoxins of all kinds of feed materials and products, the lowest levels are all below the detection levels. The maximum could be an indicator of mycotoxin contamination degrees. The positive rates and the positive averages could be indicators of the prevalence and the average contamination levels of the mycotoxins. The system has been well established. So we remained the original table contents for now, if I didn’t get your suggestions right, please let me know and I will revise accordingly.
Point 7: same comment as above - also there is a typo in the table description - "materials"
Respond 7: The typo has been modified. Please see the revised Table 3.
Point 8: Also in the table would it be a good idea to indicate what the positive threshold for mycotoxins are in China, so it is easy to assess the level of contamination. When the authors say positive do they mean over this threshold or do they simply mean detected at any level above the limit of detection of the technique?
Respond 8: In this manuscript, we defined the samples as positive if the values of concentrations are higher than the threshold of mycotoxin concentrations as listed below: > 1 μg/kg for the sum of AFB1, AFB2, AFG1 and AFG2, > 32 μg/kg for ZEN; > 50 μg/kg for trichothecenes type B, and > 100 μg/kg for the sum of FB1, FB2 and FB3. Because the mycotoxin survey has lasted for relatively long time, we set the threshold values for the consistency of LODs of different detection methods. Please see Line 521—523.
Point 9: Just a general question but why do the authors feel that Aflatoxins are much lower incidence than the other mycotoxins - is this just due to the difference in levels making low level Aflatoxin contamination difficult to detect?
Respond 9: Aflatoxins are much less prevalent probably because in China, with the temperate and sub-tropical climatic conditions, aflatoxins do not occur as common as that in tropical regions. This is my short view. Please give me some other suggestions if you have further opinions.
Point 10: For Figure 1 the geographic location and the discussion of the results is it possible to link these differences to weather etc.
Respond 10: In the discussion section, we’ve roughly discussed the possible relations of mycotoxin contamination levels with weather conditions. At the same time, we also relized that there are lots of factors influencing the mycotoxin contamination, while the temperatures and precipitation might play important roles. For now, there is already a project ongoing launched by our company to investigate the relations of weather conditions, mainly the temperature and precipitation, and mycotoxin concentrations on a global scale. This requires lots of data sources and statistical analysis and will take relatively long time to obtain a relatively accurate results of correlations. It’s meaningful and we will improve this in the follow-up work and actively publish relevant results.
Point 11: Discussion - "The large differences in mycotoxin occurrence could be contributed by the great diversity in climate conditions," This does not seem very certain are there studies that have linked climatic change to development of mycotoxins
Respond 11: In recent years, the possible impacts of climate change on mycotoxin contamination have been addressed by a number of publications. The references have been added in the text and the References part, please see Line 346 and References 29-32.

Round 2
Reviewer 1 Report
In my opinion, despite the reasonable answers of the Authors, given the considerable overall amount of data collected, a greater insight into the data is needed. This can be done, in my opinion, by following the advice regarding data analysis that I suggested in the first review, or at least by discussing it extensively, subpoint by subpoint, in the discussion. For this reason I believe that, until that the work said here above has not been made, the article will not yet have the scientific consistency necessary for publication.
Author Response
Respond 1: The statistical analysis of the raw data has been supplemented in this survey. For the general mycotoxin contamination during the 5-year period, we compare the difference of individual mycotoxin contamination based on the variation of year, and the reason that we didn’t take the type of the feed materials and products as a variation in this survey is that it has been well proved that the patterns of mycotoxin contamination in different feed commodities are different. The corresponding modifications are detailed in the revised version of Table 2, please see the revised table 2 and line 111-117. For mycotoxin contamination in new season maize from 2017 to 2021, the variation of year and region were considered to compare the difference of mycotoxin contamination in this survey. The corresponding modifications are detailed in the revised Table 4 and Table 5. And the modifications in the results description, please see line 163-167, line 187-223.
We also added the P-values of correlations of two mycotoxins in new season corn. Please see the revised version 2 of Figure 3, and line 247-248.
The specific description of the method of statistical analysis has been modified in the Materials and Methods part. Please see line 543-558.
If the statistical analysis fails to meet your requirements, please give us further guidance and suggestions. Thank you very much for your advice.
